# Myeloid-Derived Suppressor Cells (MDSCs) at the Crossroad of Senescence and Cancer

**DOI:** 10.3390/cancers17132251

**Published:** 2025-07-04

**Authors:** Giovanna Talarico, Stefania Orecchioni, Paolo Falvo, Francesco Bertolini

**Affiliations:** 1Laboratory of Hematology-Oncology, European Institute of Oncology IRCCS, Via Ripamonti 435, 20141 Milan, Italy; giovanna.talarico@ieo.it (G.T.); stefania.orecchioni@ieo.it (S.O.); 2Onco-Tech Lab, European Institute of Oncology IRCCS and Politecnico di Milano, 20141 Milan, Italy

**Keywords:** MDSC, cancer, aging, senescence

## Abstract

This review examines how myeloid-derived suppressor cells (MDSCs) might contribute to an age-related increase in cancer incidence, traditionally attributed to the accumulation of somatic DNA mutations in stem cells. We discuss the emerging and potentially complementary perspective of a progressive decline in immune system fitness with age, which reduces its ability to detect and eliminate neoplastic cells. In particular, the review discusses the evidence implicating MDSCs in the suppression of immune function and their contribution to immunosenescence. Although several aspects of MDSC activity still remain unclear and more in-depth investigation is needed, advancing this area of research may lead to novel strategies for improving cancer treatment outcomes in elderly patients.

## 1. Introduction

In the evolving landscape of cancer immunotherapy, growing evidence underscores the pivotal role of a robust and functional immune system in supporting strategies aimed at enhancing both innate and adaptive immune responses against tumor cells [1]. This is especially relevant for elderly patients, particularly those previously subjected to immune-toxic treatments such as certain chemotherapeutic agents. In spite of this and epidemiological data indicating that the large majority of cancer cases occurs after the sixth decade of life [2], most preclinical studies still rely on very young mice—typically only a few weeks old—as in vivo models for cancer research. These animals, while possessing a fully functional immune system and competent thymus [3], fail to accurately mirror the immunological environment of aged individuals.

As age advances, the thymus undergoes progressive involution, resulting in a significant reduction in T cell production and a diminished diversity and functionality of the T cell repertoire—a hallmark of immunosenescence [4]. This process is marked not only by impaired immune functions, but also by the accumulation of senescent immune and non-immune cells. Although metabolically active, these senescent cells are arrested in the cell cycle and secrete a pro-inflammatory milieu of cytokines, chemokines, and growth factors collectively known as the senescence-associated secretory phenotype (SASP) [5]. While senescence may initially act as a tumor-suppressive and tissue-repair mechanism—particularly through oncogene-induced senescence—its chronic persistence disrupts tissue homeostasis and exacerbates immune dysfunction [6].

A central mediator linking senescence, chronic inflammation, and immune suppression is the expansion of myeloid-derived suppressor cells (MDSCs). These immunosuppressive cells of myeloid origin expand in response to pathological inflammatory stimuli such as cytokines (e.g., IL-6) and colony-stimulating factors (e.g., GM-CSF) that are elevated during aging and in SASP. These factors promote MDSC mobilization and activation, particularly to sites of inflammation or damage. In aging, this contributes to the amplification of “inflammaging”—a chronic, low-grade inflammatory state described in Figure 1—and further impairs immune surveillance [7,8,9].

Taken together, these age-related immunological alterations—including thymic involution, immunosenescence, SASP-driven inflammation, and MDSC-mediated immune suppression—reveal a critical gap in cancer immunotherapy research. Addressing this gap necessitates the integration of age-appropriate, physiologically relevant models to improve therapeutic efficacy in elderly patients. In this review, we explore the role of MDSCs, focusing on preclinical models of neoplasia and cancer patients.

## 2. MDSCs in the Aged TME: Drivers of Tumor Progression and Immune Evasion

Aging is a critical factor in the incidence and progression of many cancer types, reflecting profound age-related changes in both tumors and their surrounding microenvironments [10]. With global populations rapidly aging, understanding the biological interplay between aging and cancer has become increasingly critical, as the majority of cancer diagnoses occur in older adults. Common malignancies like prostate, colorectal, lung, and pancreatic cancers are more common in older patients. Although breast cancer can occur in younger women, it is more common in older ones [11,12,13,14,15]. This increased vulnerability arises not only from the gradual accumulation of genetic mutations over time, but also from systemic alterations in the tumor microenvironment (TME), declining immune surveillance, and broader physiological changes that emerge with aging [16,17].

One key factor is thymic involution, which significantly reduces the output of naïve T cells, limiting the diversity of the T cell receptor (TCR) repertoire, a critical element for recognizing a wide array of tumor antigens. This reduction in T cell diversity, combined with functional impairments in T cell activation, makes it more challenging for the aging immune system to mount effective responses against emerging tumors [1,4]. Moreover, inflammaging is driven by a combination of cellular senescence, mitochondrial dysfunction, altered gut microbiota, and chronic antigenic load. This pro-inflammatory environment not only contributes to systemic tissue damage, but also fosters a TME that is more permissive to tumor initiation, progression, and metastasis. The release of pro-inflammatory cytokines, including IL-6, TNF-α, and IL-1β, can promote tumor cell proliferation, survival, and immune evasion, creating a vicious cycle that further compromises immune surveillance [9].

The aged TME undergoes significant structural and functional changes that contribute to cancer progression. Cancer-associated fibroblasts (CAFs) in older tissues, for instance, often exhibit a more pro-tumorigenic phenotype, characterized by reduced extracellular matrix (ECM) deposition, the increased secretion of matrix metalloproteinases (MMPs), and altered interactions with surrounding stromal and immune cells [18,19]. These changes disrupt normal tissue architecture, promote invasive tumor growth, and facilitate metastasis by degrading the ECM and releasing pro-tumorigenic signaling molecules [20]. While CAFs are classically associated with solid tumors, emerging evidence suggests that stromal fibroblast-like cells in the bone marrow microenvironment also contribute to the progression of liquid tumors including leukemias and lymphomas [21]. These cells may support malignant hematopoietic cells through cytokine production, immunosuppressive signaling, and matrix remodeling, mirroring several functions of CAFs in solid tumors [22,23]. Recognizing their role in hematologic malignancies highlights the importance of further investigating stromal contributions to immune evasion and disease progression across diverse cancer types.

In parallel, the accumulation of senescent cells in aged tissues fosters a tumor-permissive environment. These cells, characterized by SASP, secrete a complex cocktail of pro-inflammatory cytokines, chemokines, growth factors, and proteases that can disrupt local tissue homeostasis, promote chronic inflammation, and enhance tumor growth [5,24]. SASP contributes to ECM remodeling, alters tissue stiffness, and supports the recruitment of immunosuppressive cells, further undermining effective anti-tumor immunity [25,26].

One of the most impactful changes in the aged immune landscape is the increased accumulation of MDSCs, a heterogeneous population of immature myeloid cells that expand in response to chronic inflammatory conditions such as those commonly associated with aging and cancer. These cells play a pivotal role in suppressing both innate and adaptive immune responses [27,28]. MDSCs are broadly classified into two main subsets: granulocytic MDSCs (G-MDSCs), which closely resemble neutrophils in phenotype and morphology, and monocytic MDSCs (M-MDSCs), which are similar to monocytes [29]. As shown in Figure 2, the G-MDSC and M-MDSC phenotypes are characterized with different approaches in mice and humans. Within the TME, MDSCs play a central role, orchestrating a complex immunosuppressive network. This activity significantly compromises the effectiveness of various immunotherapeutic strategies including immune checkpoint inhibitors (ICIs), CAR-cells therapies, oncolytic viruses, and cancer vaccines [30,31].

Under physiological conditions, MDSCs arise from hematopoietic stem cells via myeloid progenitors. However, their differentiation into mature granulocytes, monocytes, macrophages, and dendritic cells can be disrupted under pathological states, resulting in the persistent expansion of immunosuppressive subsets [32]. Peripheral inflammatory disorders can exacerbate MDSC expansion by inducing emergency myelopoiesis, a stress-driven hematopoietic process mediated by colony-stimulating factors (CSFs) and chemokines. This process mobilizes immature myeloid cells to peripheral immune organs, such as the spleen, facilitating extramedullary myelopoiesis—a phenomenon termed demand-adapted myelopoiesis— that sustains MDSC accumulation and activation [32,33].

MDSCs suppress immune responses through both metabolic and receptor-mediated mechanisms. They produce immunosuppressive mediators such as arginase-1 (ARG1), indoleamine-2,3-dioxygenase (IDO), inducible nitric oxide synthase (iNOS), reactive oxygen species (ROS), transforming growth factor-beta (TGF-β), and interleukin-10 (IL-10). These molecules inhibit T cell proliferation and function, foster the expansion of regulatory T cells (Tregs), and induce oxidative stress and apoptosis, collectively reinforcing a profoundly immunosuppressive microenvironment [30,34].

In addition to their known expansion in aging and inflamed environments, a hypoxic TME is also a potent inducer of MDSC accumulation and activation [35]. Hypoxia upregulates the expression of hypoxia-inducible factors (HIFs), particularly HIF-1α, which in turn drives the expression of immunosuppressive mediators like vascular endothelial growth factor (VEGF), ARG1, and PD-L1, facilitating MDSC recruitment and functional polarization [36]. Once within the TME, MDSCs exert broad immunosuppressive effects including the inhibition of cytotoxic T and NK cell activity and the induction of other regulatory immune populations. Specifically, MDSCs facilitate the differentiation and expansion of Tregs through IL-10 and TGF-β [37], promote regulatory B cells (Bregs) via IL-10 [38], and contribute to the polarization of tumor-associated macrophages (TAMs) toward an M2-like, tumor-promoting phenotype through the release of S100 calcium-binding protein A8/A9 (S100A8/A9) proteins and chemokines such as C–C motif chemokine ligand 2 (CCL2). Notably, monocytic MDSCs (M-MDSCs) can differentiate into TAMs upon infiltration into the tumor site, thereby reinforcing immunosuppressive feedback loops and supporting tumor progression [39]. Importantly, TAMs derived from M-MDSCs are not merely passive components of the TME but actively sustain immunosuppression. These cells secrete high levels of IL-10 and TGF-β, inhibit antigen presentation, suppress cytotoxic T lymphocyte activity, and facilitate tissue remodeling and angiogenesis—all of which contribute to immune evasion and resistance to therapy.

Additionally, MDSCs express inhibitory ligands such as PD-L1, which engage PD-1 receptors on T cells, delivering suppressive signals that dampen T cell activity [40]. This checkpoint-mediated suppression is further exacerbated by the MDSC-driven differentiation of T cells into Tregs, perpetuating immune evasion within the TME [30].

Crucially, aging amplifies these suppressive mechanisms. The age-related decline in immune resilience exacerbates MDSC-mediated inhibition, potentially limiting the efficacy of immunotherapies in older patients. Many preclinical immunotherapeutic strategies have been developed in young animal models with fully functional immune systems, which may not adequately reflect the altered immunological landscape of aging individuals [41]. Overcoming these age-related challenges will be essential to improving cancer treatment outcomes in aging populations, calling for more tailored therapeutic approaches and patient-specific management strategies.

## 3. MDSCs-Related Evidence in Preclinical Models of Cancer and Other Diseases

As discussed above, aging is associated with a significant remodeling of the immune cell repertoire, marked by both quantitative and functional alterations in innate and adaptive immune cell populations. This immunological remodeling underlies the increased susceptibility of elderly individuals to infections, cancer, and autoimmune disorders as well as their reduced responsiveness to vaccines and immunotherapies.

MDSCs were first described by Gabrilovich and colleagues nearly two decades ago [27,28,29]. Initially, their function was not fully understood, although they were recognized as potent regulators of immune responses. In murine models, MDSCs have been broadly categorized into two main subsets: polymorphonuclear or granulocytic MDSCs (G-MDSCs; CD11b^+^Ly6G^+^Ly6C^low^) and monocytic MDSCs (M-MDSCs; CD11b^+^Ly6G^−^Ly6C^high^). In humans, a third subset has been identified, referred to as early-stage MDSCs (eMDSCs) [42], which are defined based on their colony-forming activity. Over the past 20 years, substantial evidence has accumulated showing that MDSCs increase in various tissues of aged mice, and this phenomenon has also been observed in humans over the age of 60 [43]. Elevated MDSC levels in aging have been linked to various pathological conditions including cardiovascular dysfunction such as heart failure and ischemic stroke as well as tumor development.

In this context, Ning-Sun and colleagues recently demonstrated that aged mouse hearts exhibit a higher proportion of MDSCs, significantly increasing the risk of stroke. When MDSCs from old mice were transferred into young mice, the recipients developed cardiac fibrosis, displaying a phenotype similar to that observed in aged animals. These cells exert their effects by activating fibroblasts through the secretion of proteins S100A8 and S100A9 and by modulating a molecular pathway involving fibroblast growth factor 2 and the transcription factor SOX9. This signaling axis enables fibroblasts to evade senescence and apoptosis. These findings suggest that targeting these immune cells or their signaling mechanisms may offer a promising therapeutic strategy for treating heart failure [44].

More recently, Yan and colleagues have also correlated MDSCs with ischemic stroke development. In detail, the study examined how G-MDSCs and treatment with anti-Ly6G antibodies influenced ischemic stroke in mouse models and proposed that the preventive use of anti-Ly6G antibodies could protect against ischemic stroke by limiting the infiltration of activated neutrophils, which are part of the G-MDSC population. These results underscore the promising therapeutic potential of targeting G-MDSCs in ischemic stroke management [45].

In the context of cancer, numerous preclinical studies have demonstrated a strong correlation between MDSC expansion and increased tumor risk. In a landmark study, Grizzle et al. demonstrated in a murine model of mammary adenocarcinoma that functional T cells delayed tumor onset in young mice (2 months old). However, this anti-tumor effect was lost in aged mice (12 months old), in which reduced T cell numbers coincided with a marked expansion of splenic MDSCs. Notably, the depletion of CD11b^+^Gr1^+^ cells in aged mice resulted in tumor regression, strongly suggesting that MDSCs are key mediators of immune suppression in the tumor microenvironment [46]. Similarly, Chen et al. reported analogous findings in a murine model of lung cancer, where aged mice exhibited elevated MDSC levels, impaired T cell responses, and accelerated tumor growth. Furthermore, MDSCs in these aged mice overexpressed PD-L1, a key immune checkpoint molecule. In vivo PD-L1 blockade reduced MDSC-mediated immunosuppression and restored anti-tumor T cell activity, ultimately resulting in tumor regression [47].

We have recently investigated some facets of the role of MDSCs in cancer progression. Initial work by Orecchioni [48] and colleagues demonstrated in mouse models of B cell lymphoma that the combination of two chemotherapeutic agents—vinorelbine and cyclophosphamide—with anti-PD-L1 therapy reduced monocytic M-MDSCs and tumor burden. Building on this, subsequent studies by Falvo and colleagues reported that the combinatorial treatment with cyclophosphamide, vinorelbine, and anti-PD-1 significantly reduced tumor development in two preclinical models of triple-negative breast cancer (TNBC) [49,50,51]. Using cytometry and single-cell RNA sequencing (scRNA-Seq), we found that this therapeutic combination profoundly reshapes both the circulating and intra-tumoral immune landscapes. This immune remodeling appears to block local and metastatic tumor progression by promoting the expansion of stem cell-like anti-tumor T cells (scTs) [49,51,52]. Our earlier data support a mechanistic model in which vinorelbine activates antigen-presenting cells (APCs), while cyclophosphamide and anti-PD-1 modulate scT cell clonality and activation, respectively. This results in a robust and sustained anti-tumor immune response. Similar results were also observed in a model of non-Hodgkin lymphoma (NHL), demonstrating in addition that an endogenous immune system protects tumor development upon tumor re-challenge [52]. This innovative immunomodulatory strategy aims to reprogram the immune system to combat the tumor, shifting the focus away from direct tumor targeting.

We also recently demonstrated that this therapeutic effect on TNBC growth was maintained across different ages in preclinical models—both in young mice (6–8 weeks, analogous to puberty in humans) and in adult mice (12 months, comparable to 40-year-old humans, the typical age of TNBC onset). These findings suggest that an aging immune system does not compromise the therapy’s anti-tumor efficacy [41]. We actually analyzed the TME composition by scRNA-Seq, showing that several differences were present in the immune landscape in old and young mice such as an increase in NK cells in the young mice, and in B lymphocytes in the adult ones. However, in both age groups, we observed that the treatment consistently reduced the MDSC levels, thereby enhancing T cell-mediated anti-tumor activity. Moreover, we observed that all markers for antitumoral T stem cells were maintained, enhancing anti-tumor activity.

## 4. Myeloid-Derived Suppressor Cells in Elderly Cancer Patients

The progressive increase in global life expectancy has contributed to a demographic transition in which cancer is now predominantly a disease of older adults, with over half of all malignancies diagnosed in individuals aged 65 years and above. Aging is associated with profound alterations in immune function, collectively known as immunosenescence, which contribute to the increased susceptibility to infections, cancer, and autoimmune conditions observed in older adults. This phenomenon includes both adaptive and innate immune alterations. On the adaptive side, thymic involution and the reduced output of naïve T cells result in a skewed and oligoclonal T cell repertoire [53]. Similarly, B cell diversity declines, impairing antigen-specific responses [54,55]. On the innate side, although myeloid compartments may appear numerically preserved or even expanded, they exhibit functional impairments and dysregulated responses [56,57,58].

These immunological shifts are accompanied by inflammaging, characterized by elevated levels of pro-inflammatory cytokines, C-reactive protein, and blood coagulation factors [59]. This persistent inflammation is not only a hallmark of aging, but also a significant driver of tissue damage, immunosuppression, and tumorigenesis in the elderly.

The aging process leads to a profound remodeling of the hematopoietic system, where hematopoietic stem cells (HSCs)—the progenitors of both adaptive and innate immune cells—undergo functional decline characterized by reduced regenerative capacity, increased DNA damage, and impaired self-renewal [60,61]. A hallmark of this age-related dysfunction is a shift in lineage commitment, with a pronounced myeloid bias favoring the production of monocytes, neutrophils, and dendritic cells at the expense of lymphoid lineages such as T cells, B cells, and natural killer (NK) cells.

This myeloid skewing is driven by multiple age-related factors. One major contributor is clonal hematopoiesis of indeterminate potential (CHIP), a condition that becomes increasingly common with age and is characterized by somatic mutations—most frequently in genes like *DNMT3A* and *TET2*—that promote myeloid proliferation and are associated with chronic inflammation and heightened cancer risk [62]. Another contributing factor is inflammaging itself: the sustained, low-grade production of inflammatory cytokines such as IL-6, IL-1β, and TNF-α fosters a microenvironment that promotes emergency myelopoiesis, favoring the expansion of immature and immunosuppressive myeloid progenitors [63]. Finally, structural and functional changes in the aged bone marrow niche—including impaired stromal support, decreased osteoblast activity, and increased marrow adiposity—disrupt hematopoietic regulation and further support the generation of immunosuppressive populations like MDSCs [64].

HSCs committed to the myeloid lineage are more prevalent than lymphoid progenitors in humans [65]. This skewed myeloid/lymphoid ratio in humans is further supported by the upregulation of genes associated with myeloid lineage commitment such as GM-CSF signaling as well as genes linked to malignancies including AURKA, FOS, and MYC. Conversely, genes critical for lymphoid specificity and function, such as FLT3 and SOX4, are downregulated [65].

This skewing of hematopoiesis promotes the accumulation of immature myeloid cells, including MDSCs, driven by age-related changes in the bone marrow microenvironment and pro-inflammatory signaling pathways such as the IL-6/STAT3 axis. These molecular cues favor the expansion and persistence of MDSCs, which in turn amplify immunosuppressive circuits and tumor-promoting inflammation. MDSC subsets employ distinct suppressive mechanisms, with M-MDSCs using arginase, inducible nitric oxide synthase (iNOS), and immunosuppressive cytokines, and G-MDSCs primarily relying on reactive oxygen species [66].

Evidence suggests that aging itself promotes the expansion of circulating MDSCs [67,68]. Verschoor et al. demonstrated that CD11b^+^CD15^+^ granulocytic MDSCs are significantly increased in the blood of community-dwelling seniors (61–76 years old), especially among the frail elderly (67–99 years) and those with a history of cancer, whereas M-MDSCs remain relatively unchanged [68]. This age-related expansion of MDSCs is accompanied by elevated systemic levels of pro-inflammatory cytokines such as TNF-α, IL-6, and IL-1β, while the frequency of the monocytic subset remains largely unchanged [68]. Similarly, Alves and colleagues found that in individuals aged 80–100, there was a significant predominance of granulocytic MDSCs, whose percentage was markedly increased, whereas the monocytic subset—significantly higher in younger individuals (20–30 years)—remained largely unaffected in the elderly [67].

Functionally, aged MDSCs show altered differentiation and function, such as reduced ability to mature and increased suppressive abilities, which are further modulated by cytokine-driven signaling pathways, including TNFR2/JNK [69], disrupting cell polarity.

Despite the fact that more than half of all cancer patients are aged 65 and older, studies specifically addressing the role of MDSCs in this population remain scarce. Most data come from malignancies prevalent in high-income countries—lung, colorectal, prostate, and breast cancers—where MDSCs have been implicated in disease progression and therapy resistance. In non-small cell lung cancer (NSCLC), elevated MDSC levels correlate with advanced disease, brain metastases, and poor outcomes. However, therapeutic responses often correlate with reductions in MDSC levels. For instance, in patients aged 56–70 treated with checkpoint inhibitors, survival improved when initially high MDSC levels declined during therapy [70]. Similar findings have been reported in elderly patients undergoing chemo-immunotherapy or stereotactic radiotherapy, where decreasing MDSC levels were linked to modest immune recovery and clinical benefit [71,72]. In colorectal cancer, MDSC expansion is linked to advanced tumor stage and aggressiveness, with the upregulation of genes involved in proliferation and migration [73,74]. These cells also impair the efficacy of immunotherapies; for example, in cancer vaccine trials, non-responders with high baseline MDSC levels exhibited suppressed T cell activity, which was partially reversed upon MDSC depletion, leading to restored interferon-γ production [75]. This evidence suggests that targeting MDSCs may represent a valuable strategy to enhance the efficacy of immune-based therapies, including checkpoint inhibitors and cancer vaccines, particularly in older patients. In prostate cancer, especially in older individuals with castration-resistant disease, both G- and M-MDSCs are elevated and associated with poor prognostic indicators like high PSA levels and reduced overall survival [76,77]. Importantly, MDSC reduction in the context of cancer vaccines or immune checkpoint blockade has been linked to improved clinical outcomes [78,79]. Similarly, in breast cancer, MDSCs sustain an immunosuppressive environment even during therapy. While certain treatments may initially elevate the M-MDSC levels, subsequent declines have been correlated with the recovery of T and NK cell function and improved patient responses [80,81]. A summary of age-related MDSC features, their clinical associations, and potential therapeutic strategies is provided in Table 1.

Notably, the accumulation of MDSCs in elderly patients contributes to treatment resistance by sustaining an immunosuppressive microenvironment and dampening the efficacy of immunotherapies. Reductions in MDSC levels during therapy have been associated with improved immune responses and clinical outcomes, highlighting their role as potential modulators of therapeutic success.

Altogether, understanding the intersection between immunosenescence and MDSC biology is critical for developing effective, age-tailored cancer therapies. A deeper mechanistic understanding of how aging shapes MDSC generation, phenotype, and function in elderly individuals with cancer is urgently needed to identify novel therapeutic vulnerabilities and improve patient outcomes.

## 5. Conclusions

Current paradigms associate the dramatic increase in cancer incidence after the sixth decade of human life to the accumulation of age-related, pathological, DNA somatic variants in the stem cell compartment of the affected tissue [2]. An alternate, possibly complementary, field of novel investigations focuses on the senescence-related decrease in fitness of the immune system, that in the first part of human life seems to be more efficient in patrolling and controlling neoplastic cells and cancer development. As discussed here, the role of MDSCs in dampening the immune orchestra and in age-related immune fitness decrease is emerging, but many facets have not been fully elucidated, with particular reference to the need for more preclinical and clinical studies comparing the role of MDSCs in aging vs. young models and patients. Notably, this field of research has the potential to overcome some age-related barriers in the treatment of elderly cancer patients. In fact, a more complete understanding of the fitness of the immune cell orchestra in elderly vs. mid-aged cancer patients might help to tailor cancer treatments to the patient’s overall health rather than just chronological age. Along this line, researchers and regulatory agencies should encourage trial designs that include older adults, and data should be evaluated specifically for older patients to guide evidence-based care.

## Figures and Tables

**Figure 1 cancers-17-02251-f001:**
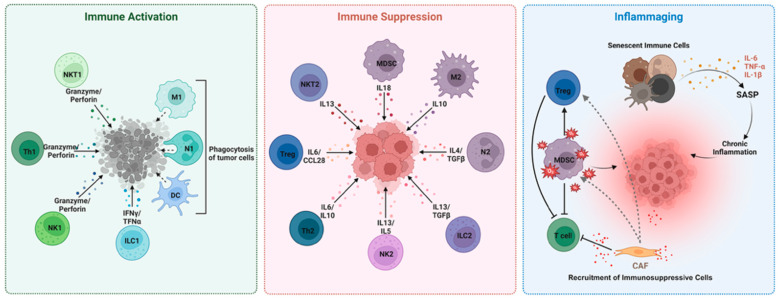
The image illustrates the transition of immune responses from anti-tumor activation to immunosuppressive and dysregulated states driven by aging and chronic inflammation. The left panel shows type 1 immune responses (e.g., NK1, Th1, ILC1, NKT1, M1 macrophages, N1 neutrophils, DCs) promoting tumor clearance via cytotoxic molecules (granzyme, perforin, IFN-γ, TNF-α), phagocytosis, and cell killing. The central panel highlights type 2 and regulatory cells (e.g., NK2, Th2, ILC2, NKT2, M2 macrophages, N2 neutrophils, MDSCs, Tregs) that secrete IL-4, IL-10, IL-13, and TGF-β to suppress anti-tumor responses and support tumor growth. The right panel illustrates how senescent immune and stromal cells, including cancer-associated fibroblasts (CAFs), drive chronic inflammation via SASP factors (e.g., IL-6, IL-1β, TNF-α, ROS), matrix remodeling, and recruitment of immunosuppressive cells, ultimately impairing immune surveillance and fostering tumor progression in aged tissues. Figure was created with Biorender software (https://www.biorender.com/ accessed on 2 July 2025).

**Figure 2 cancers-17-02251-f002:**
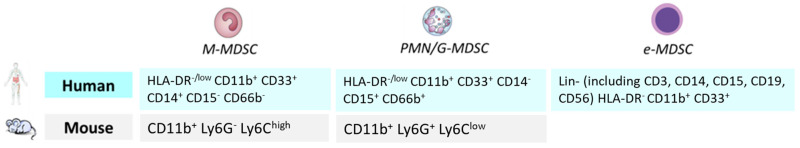
M-MDSC and PMN/G-MDSC phenotypes in mice and humans and e-MDSC phenotype in humans.

**Table 1 cancers-17-02251-t001:** Summary of age-related MDSC features, clinical associations, and potential therapeutic strategies.

Feature/Observation	Details	Clinical Implication	Potential Strategy
MDSC expansion with age	Predominantly granulocytic MDSCs (G-MDSCs) increase in elderly; M-MDSCs remain stable	Associated with impaired anti-tumor immunity	Targeted depletion of G-MDSCs; monitoring as a biomarker
Functional enhancement	Increased suppressive capacity; altered differentiation; cytokine-driven polarization (e.g., via TNFR2/JNK pathway)	Promotes immune evasion and resistance to immunotherapy	Inhibition of suppressive signaling pathways (e.g., STAT3, TNFR2)
Inflammaging and emergency myelopoiesis	Chronic low-grade inflammation (IL-6, IL-1β, TNF-α) favors expansion of immature myeloid cells	Fuels MDSC accumulation and tumor-supportive microenvironment	Anti-inflammatory therapies; inhibitors of IL-6/STAT3 axis
Clonal hematopoiesis and myeloid bias	CHIP (e.g., TET2, DNMT3A mutations) and bone marrow remodeling drive myeloid skewing	Sustained MDSC output from progenitor pools	Targeting CHIP-related pathways; niche-modifying agents
Tumor-specific associations	High MDSCs correlate with poor outcomes in lung, colorectal, prostate, and breast cancers	Predictive of resistance to chemo- and immunotherapy	MDSC monitoring to stratify patients; combination therapies
Immunotherapy interactions	MDSC reduction linked to improved response to ICIs and cancer vaccines	MDSCs limit efficacy of T and NK cell-based therapies	Combine immunotherapies with MDSC-targeting agents
Age-related clinical gap	Elderly underrepresented in trials; limited age-specific data on MDSCs	Limits generalizability of immunotherapy outcomes	Inclusion of elderly in clinical trials; development of geriatric immuno-oncology models

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
