# Peer review of "Myeloid-Derived Suppressor Cells (MDSCs) at the Crossroad of Senescence and Cancer"

_cancers, 2025, doi:10.3390/cancers17132251_

Round 1
Reviewer 1 Report
Comments and Suggestions for Authors
MDSCs are known to accumulate in both cancer and senescent tissues, where they suppress immune surveillance and facilitate disease progression. Exploring their function at the intersection of cellular senescence and tumor development may reveal shared mechanisms that drive immune evasion, chronic inflammation, and tissue dysfunction. This line of investigation holds promise for identifying novel therapeutic targets that could simultaneously address tumor growth and the detrimental effects of aging on the immune system, thereby contributing to the development of more effective immunotherapies and interventions for age-associated disease, especially cancer.
The review is both timely and highly relevant, as it explores a critical interface between two major biological processes with significant clinical implications. It provides valuable insights that contribute to our understanding of how these cells influence cancer development in the elderly population. Given the growing interest in targeting the tumor microenvironment and age-related immune decline, this comprehensive synthesis offers a meaningful contribution to the field. Therefore, I recommend the review for publication.
Minor remarks:
Please provide the full names of the abbreviations the first time they appear in the text and below the figures (figure captions).
Line 73: replace „cartoon” with „image”, „graph”, or „illustration.” Also, please add an abbreviation list in the figure description.
Author Response
|
Response to Reviewers
|
||
|
1. Summary |
|
|
|
Dear Editor and dear Reviewers, thank you very much for taking the time to review this manuscript, for constructive criticism and useful comments. Please find the detailed responses below and the corresponding revisions/corrections highlighted/in track changes in the re-submitted files.
|
||
|
2. Point-by-point response to Comments and Suggestions for Authors
Reviewer #1 |
||
|
Comment 1: Please provide the full names of the abbreviations the first time they appear in the text and below the figures (figure captions).
|
||
|
Response 1: Thank you for pointing this out. We have added the full list of abbreviations at the end of the manuscript after the “conflict of interest” paragraph.
|
||
|
Comment 2: Line 73: replace „cartoon” with „image”, „graph”, or „illustration.” Also, please add an abbreviation list in the figure description.
Response 2: Modified as requested |
||
Reviewer 2 Report
Comments and Suggestions for Authors
The review by Talarico et al. offers valuable insights into the immunosuppressive roles of myeloid-derived suppressor cells (MDSCs) and their interactions within the tumor microenvironment (TME) in elderly patients. MDSCs play a central role in tumor progression and immune evasion, making them a significant barrier to effective cancer therapy. These cells accumulate in the tumor microenvironment (TME) in response to chronic inflammation and are potent inhibitors of antitumor immunity. Given their crucial role in immune resistance, MDSCs represent a compelling therapeutic target, with ongoing efforts focused on depleting their numbers, inhibiting their function, or preventing their recruitment to the TME. The review is well-written and effectively establishes the major role of MDSCs in elderly patients with cancer. However, some minor revisions are necessary to enhance the manuscript's quality before publication.
In line 60, the authors discuss the expansion of myeloid-derived suppressor cells (MDSCs) in response to "pathological inflammatory stimuli." They should provide specific examples of such stimuli, such as cytokines like IL-6 and GM-CSF, which are known to drive the accumulation of MDSCs.
- In line 108, the authors mention cancer-associated fibroblasts (CAFs). Since CAFs are traditionally associated with solid tumor stroma, it would be valuable to discuss their role in liquid tumors as well, enhancing the manuscript’s comprehensiveness. Additionally, the authors should address how the hypoxic tumor microenvironment promotes the accumulation of MDSCs and note that MDSCs contribute to the development of various immunosuppressive cells, including regulatory T cells (Tregs), regulatory B cells (Bregs), and tumor-associated macrophages (TAMs). Including the insight that monocyte-derived MDSCs (M-MDSCs) can differentiate into tumor-associated macrophages (TAMs) upon infiltration into the tumor microenvironment (TME) would be beneficial. It is essential to emphasize that TAMs derived from M-MDSCs exhibit immunosuppressive properties.
- The authors should expand their discussion to address the role of MDSC accumulation in treatment resistance explicitly. While the immunosuppressive functions of MDSCs are acknowledged, their direct impact on reducing the efficacy of cancer therapies, especially immunotherapy, chemotherapy, and radiotherapy, requires further exploration.
- CAFs are first mentioned in Figure 1 (line 80) without prior definition, while their full term appears later in line 109. A comprehensive list of abbreviations is recommended for clarity and consistency.
- The term "myeloid-derived suppressor cells" (MDSCs) is defined in line 59 but is redundantly redefined in line 287. The authors should remove the second definition.
- The citations for Chen et al. (line 209) and Grissle et al. (line 204) need to be checked for consistent formatting.
Author Response
Reviewer #2
Comment 1: In line 60, the authors discuss the expansion of myeloid-derived suppressor cells (MDSCs) in response to "pathological inflammatory stimuli." They should provide specific examples of such stimuli, such as cytokines like IL-6 and GM-CSF, which are known to drive the accumulation of MDSCs.
Response 1: Thank you for pointing this out. We agree with this comment. Therefore, we have addressed the reviewer’s comment at line 60 of the revised manuscript: ”.. These immunosuppressive cells, of myeloid origin, expand in response to pathological inflammatory stimuli such as cytokines (e.g., IL-6) and colony-stimulating factors (e.g., GM-CSF), which are elevated during aging and in SASP. These factors promote MDSC mobilization and activation, particularly to sites of inflammation or damage. In aging, this contributes to the amplification of "inflammaging"—a chronic, low-grade inflammatory state described in Figure 1—and further impairs immune surveillance”.
Comment 2: In line 108, the authors mention cancer-associated fibroblasts (CAFs). Since CAFs are traditionally associated with solid tumor stroma, it would be valuable to discuss their role in liquid tumors as well, enhancing the manuscript’s comprehensiveness. Additionally, the authors should address how the hypoxic tumor microenvironment promotes the accumulation of MDSCs and note that MDSCs contribute to the development of various immunosuppressive cells, including regulatory T cells (Tregs), regulatory B cells (Bregs), and tumor-associated macrophages (TAMs). Including the insight that monocyte-derived MDSCs (M-MDSCs) can differentiate into tumor-associated macrophages (TAMs) upon infiltration into the tumor microenvironment (TME) would be beneficial. It is essential to emphasize that TAMs derived from M-MDSCs exhibit immunosuppressive properties.
Response 2: We thank the reviewer for this insightful comment.
Firstly, regarding CAF role and function we have fully addressed reviewer’s comment in lines 116-123.
Secondly, regarding the role of hypoxic TME and the roles of immunosuppressive cells (T regs/TAMS etc) we have fully addressed reviewer’s comment in the main text the lines 161-179.
Comment 3: The authors should expand their discussion to address the role of MDSC accumulation in treatment resistance explicitly. While the immunosuppressive functions of MDSCs are acknowledged, their direct impact on reducing the efficacy of cancer therapies, especially immunotherapy, chemotherapy, and radiotherapy, requires further exploration.
Response 3: We thank the reviewer for this observation. We have added the lines 363-367.
Comment 4: CAFs are first mentioned in Figure 1 (line 80) without prior definition, while their full term appears later in line 109. A comprehensive list of abbreviations is recommended for clarity and consistency.
Response 4: We thank the reviewer. We have corrected line 81 by adding CAF definition. Moreover, as underlined by R1Q1 we have reported at the end of the manuscript a list of all the abbreviations.
Comment 5: The term "myeloid-derived suppressor cells" (MDSCs) is defined in line 59 but is redundantly redefined in line 287. The authors should remove the second definition.
Response 5: Thanks for pointing this out, we have corrected the definition in line 313.
Comment 6: The citations for Chen et al. (line 209) and Grissle et al. (line 204) need to be checked for consistent formatting.
Response 6: Thanks and sorry for the mistake. We have adapted all the references using MDPI style.
Reviewer 3 Report
Comments and Suggestions for Authors
The review entitled: „Myeloid-derived suppressor cells (MDSCs) at the crossroad of
senescence and cancer“(ID: cancers-3714016) by Talarico et al. explores the role of MDSC with the focus on preclinical models of neoplasia and cancer patients.
Albeit the review is well written of special interest, comments should be addressed to further improve the manuscript.
Comments:
- Conclusions: the authors should highlight more intensively how this potential clinical need could be solved. It is very important to note, that also the authors should discuss more intensively in this section how future research in this field could potentially overcome age related barriers in the treatment of elderly patients.
- For a better overview it would be helpful to provide a table where the main insights/facts from the current studies or results are combined.
- Figure 1 is too small and should be enlarged to make it more readable.
Author Response
Reviewer #3
Comment 1: Conclusions: the authors should highlight more intensively how this potential clinical need could be solved. It is very important to note, that also the authors should discuss more intensively in this section how future research in this field could potentially overcome age related barriers in the treatment of elderly patients.
Response 1: We thank the reviewer for this useful comment. We have added a sentence in the conclusion part lines 384-389.
Comments 2: For a better overview it would be helpful to provide a table where the main insights/facts from the current studies or results are combined.
Response 2: We thank the reviewer for this insightful comment. We have added table 2 reporting all the information required.
Comments 3: Figure 1 is too small and should be enlarged to make it more readable.
Response 3: We thank the reviewer for this suggestion. We have modified the figure by increasing font size and resolution.